# Knowledge and attitudes about HIV pre-exposure prophylaxis: Evidence from in-depth interviews and focus group discussions with policy makers, healthcare providers, and end-users in Lesotho

Pascal Geldsetzer[1,2‡]*, Joy J. Chebet[3‡], Tapiwa Tarumbiswa[4], Rosina Phate-Lesihla[5], Chivimbiso Maponga[6], Esther Mandara[6], Till Bärnighausen[7,8,9], Shannon A. McMahon[7,10]

1 Division of Primary Care and Population Health, Department of Medicine, Stanford University, Stanford, California, United States of America, 2 Chan Zuckerberg Biohub, San Francisco, California, United States of America, 3 Department of Health Promotion Sciences, Mel and Enid Zuckerman College of Public Health, University of Arizona, Tucson, Arizona, United States of America, 4 Disease Control Department, Lesotho Ministry of Health, Maseru, Lesotho, 5 HIV Program, Lesotho Ministry of Health, Maseru, Lesotho, 6 Clinton Health Access Initiative–Lesotho Country Office, Maseru, Lesotho, 7 Heidelberg Institute of Global Health, Heidelberg University, Heidelberg, Germany, 8 Department of Global Health and Population, Harvard T.H. Chan School of Public Health, Boston, Massachusetts, United States of America, 9 Africa Health Research Institute (AHRI), Durban, South Africa, 10 Social and Behavioral Interventions, Johns Hopkins Bloomberg School of Public Health, Baltimore, Maryland, United States of America

‡ PG and JJC are joint first authors on this work.
* pgeldsetzer@stanford.edu

## Abstract

Studies on knowledge and attitudes about HIV pre-exposure prophylaxis (PrEP) have mostly focused on key populations in North America and Europe. To inform Lesotho's national rollout of PrEP to the general population, this study aimed to characterize knowledge and attitudes about PrEP among policy makers, implementing partners, healthcare providers, and PrEP end-users in Lesotho. Respondents were purposively selected to participate based on personal experience in the development and implementation of Lesotho's PrEP program, or the personal use of PrEP. We conducted 106 in-depth interviews with policy makers (n = 5), implementing partners (n = 4), and end-users (current PrEP users = 55; former PrEP users = 36; and PrEP "decliners" = 6). In addition, we held 11 focus group discussions (FGDs) with a total of 105 healthcare providers. Interview and FGD transcripts were analyzed following the tenets of Grounded Theory. Respondents expressed positive attitudes toward PrEP, owing to experienced and perceived personal, familial, and societal benefits. PrEP was viewed as i) an opportunity for serodiscordant couples to remain together, ii) a means of conceiving children with minimized risk of HIV infection, iii) providing a sense of agency and control, and iv) an avenue for addressing the HIV epidemic in Lesotho. Respondents demonstrated understanding of PrEP's intended use, eligibility requirements, and modality of use. However, respondents also reported that several important misconceptions of PrEP were common among adults in Lesotho, including a belief that

**Data Availability Statement:** The data that support the findings of this study are not publicly available due to the ethics consent form indicating that participant data would not be shared outside of the research team. These data may contain information that could compromise the privacy of research participants, and thus will not be made publicly available. This research study was approved by the Lesotho Ministry of Health ethical committee (ID03-2019) and Heidelberg University (S-865/2018). If you have any questions or concerns or requests for data access, please contact the ethics commission at the Medical Faculty of Heidelberg University at ethikkommission-I@med.uni-heidelberg.de.

**Funding:** This study was supported by the Alexander-von-Humboldt Foundation University Professorship to TB (no grant number; https://www.humboldt-foundation.de/en/). PG is a Chan Zuckerberg Biohub investigator (no grant number; https://www.czbiohub.org). The funders had no role in study design, data collection and analysis, decision to publish, or preparation of the manuscript.

**Competing interests:** The authors have declared that no competing interests exist.

PrEP protects against sexually transmitted infections other than HIV, promotes promiscuity, prevents pregnancy, causes seroconversion, and provides lifelong protection from taking the pill just once. In addition to building on the perceived advantages of PrEP to shape a positive message, Lesotho's national rollout of PrEP will likely benefit from a communication strategy that specifically addresses the common misconceptions of PrEP identified in this study.

## Introduction

Sub-Saharan Africa bears a disproportionately high HIV/AIDS burden, with approximately two-thirds of both persons living with HIV and global new infections reported in the region [1]. Behavioral HIV prevention interventions have demonstrated limited success in changing risky sexual practice, thereby suggesting a need for integrated and innovative approaches to HIV control [2]. As a means of reducing HIV acquisition, the World Health Organization (WHO) endorsed an oral prophylactic regimen for use by HIV-negative individuals in 2015 [3]. Oral Pre-Exposure Prophylaxis (PrEP), a user-initiated once-daily pill, has shown a clinical efficacy of over 90% when taken consistently [4–7]. Communities with endemic levels of HIV, and persons at greatest risk of HIV, stand to benefit most significantly from the use of PrEP [4].

The Kingdom of Lesotho reports among the highest HIV prevalence estimates globally in its general adult population (21%), with an estimated 280,000 people living with the disease in 2020 [1]. In an effort to address the national HIV/AIDS burden, the country's Ministry of Health has employed a dual approach, involving provision of care and treatment to individuals living with HIV, while concurrently implementing prevention programs targeted at HIV-negative individuals [8]. Incorporation of PrEP into Lesotho's national HIV/AIDS-related policies and strategies began in 2016, following the formation of a technical working group tasked with developing national guidelines on the use of this prophylactic [8]. A phased PrEP implementation strategy was used, with five of the ten Lesotho districts selected for the first stage, which included PrEP delivery through both facility- and community-based models. This first phase focused predominantly on providing PrEP to the HIV-negative partner in serodiscordant couples [8]. PrEP scale up to the remaining districts began in 2019. In addition to scaling up PrEP services geographically, this next phase expanded eligibility for PrEP to all individuals who are at risk of HIV and interested in taking PrEP [8].

Studies on PrEP awareness have been predominantly conducted in high-income countries [9–15], and largely among Men who have Sex with Men (MSMs) [10, 11, 13, 15–18] and healthcare providers [12, 19, 20]. Research in sub-Saharan Africa done in this area has mostly been carried out in the context of large-scale clinical trials [21–23]. There is, thus, need for additional evidence on PrEP awareness, knowledge, and perceptions from effectiveness and implementation studies conducted as part of large-scale PrEP programming as more sub-Saharan African countries incorporate PrEP into their respective national HIV/AIDS control strategies. In going beyond clinical efficacy and moving towards effective delivery of PrEP to general populations, this understanding could prove useful in allowing for development and adaptation of messaging to address misgivings and apprehensions towards PrEP, providing a launching point for course correction for incorrect/misleading information, and generating information for decision-making and resource allocation for the scale up process [24].

Our research was conducted as part of a larger implementation study, whose goal was to gain a deeper understanding of the ongoing Lesotho PrEP program and scale up process across

the health system. Focusing on policymakers, implementing partners, healthcare providers, and PrEP end-users, this present study aimed to characterize i) the current level of PrEP knowledge, ii) gaps in PrEP knowledge and misconceptions about PrEP, and iii) attitudes towards PrEP.

## Methods

### Study design, data collection activities, and participant recruitment

Through discussions with the Lesotho Ministry of Health, we identified four key stakeholder groups that we hypothesized determine the success of PrEP implementation for the general population in Lesotho: i) policy makers, ii) implementing partners, iii) healthcare providers, and iv) end-users. End-users were further divided into current PrEP users, former PrEP users, and those who were offered PrEP but declined ("PrEP decliners"). Snowball sampling informed later data collection, as a means of further expanding respondent groups.

In this qualitative study, we conducted semi-structured in-depth interviews (IDIs) with policy makers and implementing partners–including national and district level government employees, and individuals in leadership positions at funding and non-governmental organizations–to create a national backdrop which was used to contextualize data generated by other respondent groups. Key informants were identified in collaboration with the Ministry of Health, who were aware of individuals directly involved in the implementation of the national PrEP program. Respondents were purposively selected for participation in this study based on direct involvement and/or decision making in the PrEP policy development, rollout and/or implementation processes.

We conducted in-depth semi-structured interviews with adult PrEP end-users–current and former users, and decliners–to elicit detailed narratives and firsthand accounts of individual experience with PrEP (Table 1). Current PrEP users were eligible to join the study if they were actively using PrEP at the time of the interview, regardless of duration of use and/or previous non-adherence. Former PrEP users were defined as individuals who had previously used PrEP, for any period of time, but stopped using the drug, for any reason. Decliners were defined as individuals who were offered PrEP by a healthcare provider but declined to initiate PrEP. All PrEP end-users were identified through facility records and community organizations working with persons at high risk for HIV infection. To maintain client privacy, healthcare providers at the study's healthcare facilities were requested to contact and inform potential respondents about the study without revealing their client's identity. To ensure wide and representative inclusion of healthcare facilities, facilities were selected for the recruitment of study participants broadly based on: 1) type of health facility: including government, private and faith-based facilities; 2) mode of PrEP delivery: community- and facility-based; and 3)

**Table 1. Number of semi-structured in-depth interviews and focus group discussions conducted.**

| Type of data collection activities | Respondent groups | Total (n) |
|---|---|---|
| In-depth semi-structured interviews | Policy makers | 5 |
| | Implementing partners | 4 |
| In-depth semi-structured interviews | Current PrEP users | 55 |
| | Former PrEP users | 36 |
| | PrEP decliners | 6 |
| Semi-structured focus group discussions | Healthcare providers | 11 FGDs; 105 total participants |

Abbreviations: FGDs = focus group discussions; PrEP = HIV pre-exposure prophylaxis

location: urban and rural. To maximize sample size, the volume of PrEP clients at each health facility was considered, with facilities reporting the highest levels of PrEP enrollment prioritized. We selected two healthcare facilities in each of the five intervention districts. Interested PrEP end-users were invited to a central location of their choosing to learn more about the study and consented if willing to participate. Interviews were conducted in a private location of the respondent's choosing, and at a time convenient for them. To achieve diversity of perspectives and opinions, PrEP end-users were selected to represent varying ages, occupations, educational attainment, and duration of PrEP use/non-use.

We conducted semi-structured focus group discussions (FGDs) with frontline healthcare providers to gain their perspectives on the use of PrEP for prevention of HIV infection. Given the professional standing providers hold, mixed-sex FGDs were not viewed as potentially hindering open discussion and therefore men and women were invited to participate in the same FGD. Healthcare providers were eligible to participate in these discussions if they offered direct HIV and/or PrEP services. To ensure the discussions presented varied and diverse views, healthcare providers were purposively selected to reflect variation in experience, cadre, gender, and healthcare facility level.

All IDIs and FGDs were semi-structured. While the research team developed a list of pertinent questions for each respondent group, research assistants were at liberty to alter the order and sequence of research questions based on the flow of the interview or discussion (S1 Appendix). To encourage unbridled dialogue, research assistants were trained to engage in rapport building conversations prior to initiating the IDIs and FGDs. All data collection took place between March and May 2019.

## Training of interviewers and discussion moderators

All interviewers and FGD moderators had undergraduate or graduate training in sociology, gender studies, or economics. The data collection team underwent a 3-day training in qualitative research methods, research ethics, and the study procedures prior to the start of data collection. Training also included designated time for instrument pilot testing, which allowed the research team to improve data collection instruments based on feedback from the field.

## Data management

Interviews and FGDs were digitally recorded and transcribed in Sesotho and English. The interviewers took detailed notes during the interview in case of recording failure. Interviews were transcribed by the same interviewer who conducted the interview. While in the field, each interviewer was responsible for reviewing his or her demographic questionnaires for completeness and accuracy following the conclusion of an interview. As a secondary level of quality control, a study investigator reviewed all questionnaires completed each day. Demographic respondent data were entered using Microsoft Excel. Interview transcripts were managed in Atlas.ti (Scientific Software Development Gmbbh, Berlin, Germany) [25].

## Data analysis

Following each day of data collection, interviewers and FGD moderators debriefed with a lead investigator (JJC) [26]. In total, 11 debriefing sessions were undertaken. These sessions served as an opportunity to organize data, and identify emerging themes as well as areas of additional inquiry and saturated topics. Drawing on the principles of Grounded Theory, we conducted a review of academic literature of emerging themes concurrently with data collection [27]. Literature review, field notes and debriefing notes were used to develop a framework in which to analyze study transcripts. Themes were developed from detailed debriefing notes and analysis

workshops conducted between interviewers, FGD moderators, and the study investigators. We then further teased out these themes via line-by-line coding, with all codes being entered into a codebook and applied to all transcripts in Atlas.ti. [25]. The full codebook with the emerging themes (n = 3) and codes (n = 15) is available in S2 Appendix. Lastly, we summarized the data under each code and selected quotes to illustrate the theme.

### Ethics statement

Ethical approval for this research was granted by the Lesotho Ministry of Health ethical committee (ID03-2019) and Heidelberg University (S-865/2018). Written informed consent was obtained from all study participants after the interviewer read out the content of the consent form and explained the purpose of the study. Those unable to provide a signature provided a thumbprint instead. The study did not include any minors. Additional information regarding the ethical, cultural, and scientific considerations specific to inclusivity in global research is included in the Supporting Information (S1 Questionnaire).

## Results

### Respondent characteristics

On average, interviews lasted 60 minutes (SD: 19.7, range: 21–150), and FGDs lasted 77 minutes (SD: 17.3, range: 42–96). Policy makers (n = 5) and implementing partners (n = 4) included high-ranking officials at the national and district levels. On average, these policy-level respondents had been in their position for five years at the time of data collection (Table 2). Healthcare providers participating in the FGDs represented varying professional cadres, including nurses (n = 31, 30%), professional/lay counselors (n = 26; 25%), pharmacist/pharmacy technicians (n = 15; 14%), nurse assistants/student nurses (n = 11; 11%), village health workers (n = 11; 11%), and other cadres (n = 11; 11%). The healthcare providers participating in the FGDs had been in their current position for an average of 6.5 years.

PrEP end-users (current = 55; former = 36; decliners = 6) were predominantly female (79%), interviewed at an urban site (75%), and had attained some high school education (43%). Most current users (86%) were on PrEP because they were currently in a serodiscordant relationship, whereas engagement in sex work was the primary reason for PrEP use among former users (47%) and decliners (50%). On average, current and former users had been on PrEP for eight months (range: 2 days– 31 months) and four months (range: 3 days –24 months), respectively.

### PrEP knowledge

End-users had a generally high level of understanding regarding what PrEP is, how it is used and who is eligible for PrEP. End-users described PrEP as an antiretroviral drug (ARV), and named priority groups for PrEP use as, generally, individuals in serodiscordant relationships, individuals with multiple sexual partners, and MSM. When asked how they would describe PrEP in lay language to a friend, one former user said:

> "*I can tell them that PrEP is a pill that is taken by those people who are found to be [HIV] negative after running tests. It is taken every day at the same hour of the day for 20 to 28 days. You still use condoms every time [you have sex]. This builds a wall that will prevent HIV infection if it happens you meet someone who is HIV positive. Especially for sex workers like us who are always exposed to [HIV].*"

**Table 2. Sample characteristics.**

| | Policy makers and implementing partners | Healthcare providers[1] | Current users | Former users | Decliners |
|---|---|---|---|---|---|
| | *(n = 9)* | *(11 FGDs; n = 105)* | *(n = 55)* | *(n = 36)* | *(n = 6)* |
| **Age**: years; mean (SD); range | 46.7 (4.8); 41–56 | 37.3 (11.2); 20–65 | 36.4 (12.6); 20–71 | 26.7 (9.7); 18–62 | 28.8 (5.1); 22–34 |
| **Female**: n (%) | 6 (66.7) | 82 (78.1) | 38 (69.1) | 33 (91.7) | 6 (100.0) |
| **District**: n (%) | | | | | |
| Maseru | 6 (66.7) | 24 (22.9) | 7 (12.7) | 31 (86.1) | 4 (66.7) |
| Leribe | 1 (11.1) | 16 (15.2) | 12 (21.8) | 2 (5.6) | 1 (16.7) |
| Berea | 0 (0.0) | 33 (31.4) | 9 (16.4) | 1 (2.8) | 0 (0.0) |
| Mafeteng | 1 (11.1) | 18 (17.1) | 20 (36.4) | 2 (5.6) | 1 (16.7) |
| Mohales Hoek | 1 (11.1) | 14 (13.3) | 7 (12.7) | 0 (0.0) | 0 (0.0) |
| **Urban interview/FGD location**: n (%) | 9 (100) | 52 (49.5) | 32 (58.2) | 36 (100) | 5 (83.3) |
| **Years in position**[1]: mean (SD) | 5.0 (4.1) | 6.2 (6.0) | N/A | N/A | N/A |
| **Educational attainment**[2]: n (%) | | | | | |
| None or Some primary | 0 (0.0) | – | 18 (32.7) | 4 (11.4) | 0 (0.0) |
| Completed primary school | 0 (0.0) | – | 3 (5.5) | 3 (8.6) | 0 (0.0) |
| Some high school | 0 (0.0) | – | 17 (30.9) | 21 (60.0) | 4 (66.7) |
| Completed high school | 0 (0.0) | – | 10 (18.2) | 4 (5.7) | 1 (16.7) |
| Certificate/diploma | 2 (22.2) | – | 6 (10.9) | 2 (5.7) | 1 (16.7) |
| Undergraduate degree | 4 (44.4) | – | 1 (1.8) | 1 (2.9) | 0 (0.0) |
| Postgraduate degree | 3 (33.3) | – | 0 (0.0) | 0 (0.0) | 0 (0.0) |
| **Respondent HIV risk category**[3]: n (%) | | | | | |
| Serodiscordant relationship | N/A | N/A | 47 (85.5) | 5 (13.9) | 0 (0.0) |
| Migrant worker | N/A | N/A | 1 (1.8) | 0 (0.0) | 0 (0.0) |
| Partner of migrant worker | N/A | N/A | 8 (14.5) | 2 (5.6) | 0 (0.0) |
| Multiple partners | N/A | N/A | 3 (5.5) | 4 (11.1) | 0 (0.0) |
| Does not trust partner | N/A | N/A | 2 (3.6) | 4 (11.1) | 1 (16.7) |
| Female sex worker | N/A | N/A | 1 (1.8) | 17 (47.2) | 3 (50.0) |
| Pregnant/lactating woman | N/A | N/A | 3 (5.5) | 0 (0.0) | 1 (16.7) |
| Other | N/A | N/A | 0 (0.0) | 2 (5.6) | 0 (0.0) |
| **Duration on PrEP**[4]: months; mean (SD): range | N/A | N/A | 8.0 (8.0); 2 days-31 months | 4.1 (4.4); 3 days-24 months | N/A |

[1] Missing for one healthcare provider

[2] Missing for one former PrEP user

[3] Respondents could fall into more than one "risk" category

[4] Two former PrEP users and two current PrEP users were missing information on total duration on PrEP.

N/A = not applicable for this respondent group

– = these data were not collected for this respondent group

End-users reported that their first encounters with PrEP information was over the media (overwhelmingly through radio and television), education sessions at healthcare facilities (including maternal and child clinic visits, and HIV Testing and Counseling [HTC] services), and community outreach services. To a lesser degree, end-users learned about PrEP through social networks and at mobile health clinics that provided HIV services at workplaces.

Healthcare providers exhibited technical and clinical knowledge about PrEP's pharmaceutical composition and enrollment eligibility criteria. When asked about their capacity to dispense PrEP, healthcare providers expressed confidence in doing so, owing to the training they

had received. Policy makers and implementing partners explained that a Training of Trainers took place at the national level with trainers invited from each target district. These trainers were then tasked to train their colleagues at their healthcare facility. One healthcare provider explained:

> *"Well, initially when PrEP was introduced to our health facility, it was difficult especially for me to dispense it. I was always referring the clients to those who were trained first about PrEP. But lately, everyone is aware of PrEP and knowledgeable on how they can dispense PrEP."*

## PrEP misconceptions and misinformation

While PrEP end-users participating in our study were aware of PrEP, healthcare providers and policy-level respondents showed concern about the quality and accuracy of the messages that community members receive. Providers noted that when community members "*hear [about PrEP] from the streets, and not from official sources*" the information they receive is often incomplete, and in some cases, completely incorrect.

### Pregnancy and sexually transmitted infections

In addition to protecting against HIV acquisition, a few respondents reported the general public's misconception that PrEP also protects against sexually transmitted infections (STIs) and pregnancies. An implementing partner summarized this misinformation by saying *"there is a misconception that PrEP is a magic bullet. Once you are on it, you are invincible"*. A minority of our PrEP end-users themselves was also uncertain about PrEP's benefits, one asking an interviewer: "*is it true that PrEP prevents HIV infection, and also prevents unplanned pregnancies?*"

Healthcare providers postulated this misinformation arises from inaccurate information propagated by ill-trained, lay health educators. A healthcare provider elaborated on an instance they personally observed:

> "*I recall one time, I was just passing by a [health education] tent in the community, and I overheard the person talking about PrEP, telling teenagers that PrEP can stop pregnancy. That alerted me that community-based PrEP services are spreading a lot of false information, which is why we continue to encounter these issues here. I don't want to name out companies, but I have since learnt that a lot of information given at the community needs to be corrected.*"

As a result of this false impression, healthcare providers discussed concerns over lower levels of condom use leading to a higher incidence of STIs and pregnancies. A healthcare provider expressed this discomfort, noting:

> *"The main [drawback] I've observed is that that PrEP clients now think if they are using PrEP, it is no longer necessary for them to use condoms."*

### Lifelong protection

Both PrEP end-users and healthcare providers reported a misconception in the general public that a single dose of the pill confers lifelong protection from HIV. This misconception, respondents discussed, could be a result of inadequate counseling. Two current users–a woman aged 25 years from an urban setting, who had been on PrEP for 10 months, and a man aged 69

years from an urban setting, who had been on PrEP for 25 months–discussed getting the impression during counseling that only one pill is required to protected them from HIV for the rest of their lives. It was only when they received their prescription and instruction to take one pill daily that they realized that they were required to take one pill every day. Healthcare providers concurred, with one FGD participant saying:

> "*Indeed, sometimes they call it 'a pill', as if it is just one pill you can take today, and then again ten years later. So when we counsel them, it is then that they realize it is not what they initially believed it was.*"

## PrEP and seroconversion

Respondents reported that the general public often conflates the use of PrEP for HIV prevention with the use of ARVs for HIV treatment. One male current PrEP user explained, "*Many, like me in the beginning, think that PrEP and ART are the same thing*". In a minority of cases, healthcare providers explained that individuals felt like they were being "tricked" into taking antiretroviral therapy (ART), saying "*Some people think that they are being forced to take ARVs [for HIV treatment] in the form of PrEP*". Some extended this misconception, implying that the use of PrEP would cause the user to become HIV-positive. This was reflected in the experience of one current female PrEP user, whose friends told her "*Why are you inviting HIV*? *If you take PrEP, you will get HIV*". A former PrEP user explained her thought process as she was contemplating the use of PrEP, given the myth that the pill caused HIV:

> "*Many things came to my mind because [my friends] were telling me that once you start taking PrEP the next time when you get tested, your results will come out positive.*"

Healthcare providers noted that misconceptions around PrEP, ART, and becoming HIV positive could be attributed to the fact that PrEP is comprised of pharmaceutical ingredients used in ART. Additionally, this misconception could also arise from the phrasing used by healthcare providers during counseling. One current user in a serodiscordant relationship narrated that during PrEP counseling, a healthcare provider explained, "*These are your ARVs and these are his ARVs. You should not mix them. If you ran out of yours, don't take his, come to the clinic and we will give you more*", confusing the respondent and leading her to believe she too was HIV positive.

## Recommendations to improve PrEP knowledge

When discussing the origins of PrEP misconceptions prevalent at the community level, policy makers and implementing partners agreed that it is a result of poor messaging. For example, a policy maker at the national level noted that a PrEP "advertisement" on the radio says that it can be used by people who cannot use condoms consistently or if one is not sure about their partner's status. The respondent lamented that the description provided on radio was vague, and recommended more specific advertising and messaging around eligibility for PrEP. Healthcare providers tended to note the need for education and counseling programs to use trained healthcare workers who they felt are more likely to provide accurate information than lay individuals.

## Attitudes towards PrEP

Overwhelmingly, and across all respondent groups, positive attitudes were expressed towards PrEP. End-users, in particular, described the pill as "*a good thing*" and a "*gift from God*".

Respondents who knew others on PrEP or were in social circles in which PrEP use was prevalent, described PrEP as being ordinary and "*not such a big deal*". Even those who discontinued PrEP noted that although it did not work for them, they would encourage others to take PrEP.

For healthcare providers, policy makers and implementing partners, PrEP was viewed as a vehicle for meeting national HIV targets and allowing for resources to be used to manage individuals who are living with HIV. One provider noted:

> "*[PrEP] is a wonderful drug, very wonderful drug, because it helps us stop new HIV infections. So when we encourage our clients to start using PrEP, it is because we want them to remain HIV-negative for a long time, which will then allow us to just focus on our ART clients. We will have fewer new ART clients to deal with.*"

Moreover, the introduction of PrEP in Lesotho was seen as a reason for renewed hope for overcoming the long-enduring scourge of HIV in the country. One healthcare provider explained:

> "*I think it is because for so long we lived under a great fear because HIV was incurable. When HIV first became a pandemic in the country, many of us were very scared about it because of that no-cure thing. And it is true that HIV is still incurable, but nowadays that fear has largely been removed. And now that we know of these new drugs that help us to stay negative, there are new seeds of hope that one day we will become a nation without new infections.*"

For end-users, the positive attitudes conveyed about PrEP revolved around the pill's impact on intimate and familial relationships. Respondents saw PrEP as a "glue" that allows discordant couples to continue in their relationship, thus holding families together, which would have otherwise been forced apart from fear of HIV transmission. In FGDs, healthcare providers expounded on this idea, noting: "*PrEP builds families because there is no need for divorce in the case of a discordant couple. In the past, a couple would divorce if one was found to be HIV positive*". PrEP was also seen as useful in keeping families together in which there was, or continues to be, infidelity and distrust. A district-level respondent noted, "*[PrEP] has become some kind of remedy for some family disputes, which are basically being caused by lack of trust between partners*". A current user noted, "*I do not blame my wife for getting HIV. I now have a way to protect myself*".

For young couples in particular, PrEP was viewed positively as it provided an avenue for creating a family, despite having an HIV-positive partner. In fact, one current PrEP user noted that after dating her partner for many years she had not considered marriage because of concern of acquiring HIV, or not being able to conceive a child with her partner. The introduction of PrEP allowed the respondent to get married to her partner of eight years. At the time of the interview, she was pregnant with their first child.

In addition, end-users expressed a sentiment that PrEP provides a sense of safety and protection, and gives them agency and control over their reproductive health. One current user described PrEP as "*giving me life*", given the autonomy and "*peace of mind*" that PrEP offered. Healthcare providers, too, expressed the security that PrEP provides for their clients, saying:

> "*I can add that PrEP empowers women. In most cases women are hesitant to discuss sex-related issues. That being the case, they make good choices when they know their husband cheats with a neighbor. They just come and get PrEP because they know they will test HIV negative for as long as they take their medication in a proper way.*"

When asked to think of a name that they would call PrEP, many names given by end-users were reflective of the positive attitudes that they expressed towards PrEP (Table 3). They tended to provide names that evoked imagery of victory over war; an agent that conferred protection, gave renewed life, and was divine and a savior. The most common name mention, however, was one of protection, with respondents referring to PrEP as a shield (*Thebe*).

While the vast majority of respondents had positive attitudes towards PrEP, some expressed reservations about the drug. Healthcare providers in particular reported having heard negative views about PrEP as a pill that promoted promiscuity. This measured enthusiasm for PrEP is captured by one healthcare provider who noted:

"*Some see the benefits [of PrEP], others do not. Others see the benefit of protecting themselves against infection. Others see it as an encouragement for promiscuity. There are various responses to it.*"

## Discussion

Overwhelmingly, respondents expressed positive attitudes towards PrEP, owing to experienced and/or perceived personal, familial, and societal benefits attributed to PrEP. PrEP was discussed as an opportunity for couples to remain together, despite an HIV diagnosis in one partner. Furthermore, PrEP was viewed as affording serodiscordant couples the possibility to conceive children with minimized risk of HIV infection. Among respondents with limited

**Table 3. Responses by end-users when asked what name they would give PrEP.**

| War and victory imagery | Life giver | Protector |
|---|---|---|
| • *Mohloli*: to concur<br>• *Mpuli ea Mafu*: champion of disease<br>• *Sephethala*: a weapon one takes when they are going for a fight | • *Bophelo ba ka bohle*: "PrEP my life because I will need to take it forever"<br>• *Bophelo*: "life, because PrEP gives me life"<br>• *Phela*: "to live, because it gave me life. I am not afraid that I will get HIV because I am using PrEP" | • *Thebe*: shield<br>• *Sebetsa se ntsirelelitseng*: "a weapon that shields/protects me"<br>• *Sets'erelesi sa bophelo bo bottle*: protection measure against illness<br>• S*ethibela mohlako*" something that prevents pain<br>• *Makhona-tsohle*: "protects me from all corners"<br>• *Setsireletso saka*: "my shield"<br>• *Tsireletso*: shield<br>• *Seits'ereletso*: a shield<br>• *Mosereletsi oa ka*: my shield<br>• *Lerato*: "Love, because I love PrEP; it protects me" |
| **Devine/ savor/ rescuer** | **Usual/ordinary medicine** | **ARVs/HIV** |
| • *Moriana oa molimo*: medicine from God<br>• *Mopholosing*: savior<br>• *Manamolela*: "the rescuer because it rescued me from getting HIV"<br>• *mothusi oa bophelo*: my life helper/giver<br>• *Mopholosi*: the savior<br>• *Mothusioa ka*: my helper<br>• *Mothusi a moholo*: a big helper<br>• *Mampoli*: "the best of them all because it can control the new infection" | • *Panadol*: "(a pain reliever brand) because it is just a normal pill that you take every day. It is not a big deal."<br>• *Panadol*: because it is a tablet<br>• *Pilisi ea hlooho*: "headache pill because it is just like any other pill"<br>• Condom: "because it is protecting me" | • Another ARV<br>• *Pilisi e thibeleng tshoatso*: a preventative pill for HIV |

decision-making ability in sexual relationships, PrEP was seen as providing a sense of agency and control over one's HIV status, given it can be taken discreetly. Among policy makers, implementing partners, and healthcare providers, PrEP evoked a sense of hope for addressing the HIV epidemic in Lesotho. All respondent groups, however, also reported hearing or personally having initial concern that PrEP would inadvertently promote promiscuity in their community. While the vast majority of our respondents were knowledgeable about PrEP, they reported a lack of awareness in the general public. Further still, respondents reported misinformation in the general public on the benefits of PrEP, conflation with seroconversion, and the duration of the drug's efficacy.

While end-users in our study reported high levels of PrEP knowledge, it is important to note that they had all undergone counseling and testing, as part of screening and initiation on PrEP. They were, therefore, presumably more knowledgeable than the general public. All respondent groups did, in fact, report poor levels of PrEP knowledge in Lesotho generally. While other studies in the sub-Saharan region have shown poor levels of PrEP knowledge, they also indicate a strong interest in the drug once further information on PrEP is provided [28–30]. In expanding the PrEP program nationally to the general population in Lesotho, concerted sensitization efforts and community mobilization are, therefore, likely to be important to generate demand for PrEP. Furthermore, context-specific research on the level of PrEP knowledge among lay Basotho people would be useful in the scale-up and long-term implementation of a national PrEP program.

Overheard and held misconceptions around PrEP, expressed by our respondents, suggest both a misunderstanding of PrEP's use, as well as mistrust going beyond PrEP itself. Most worryingly, a misconception that PrEP protects against STIs other than HIV and pregnancy may lead an increase in unprotected sex and a higher number of sexual partners. These behavioral changes in turn may increase the incidence of STIs other than HIV and unwanted pregnancies, and also lead to HIV infections if PrEP is taken inconsistently. Concern over increases in sexual partners and promiscuity was also raised in other studies as a source of apprehension with the introduction of PrEP [31–34]. Evidence on such behavioral changes due to PrEP is mixed. Large-scale clinical trials have reported little to no such behavioral changes [4]. However, more recent smaller studies have suggested a decrease in reported condom use and increased diagnosis of STIs other than HIV among individuals on PrEP [11, 35–38]. Determining to what degree concerns about increased sexual risk-taking due to PrEP are warranted is important to be able to address this apprehension among healthcare providers who may otherwise be dissuaded from prescribing PrEP [12, 39].

In our context, the misconception that PrEP protects against STIs other than HIV and pregnancy could have arisen from a poor understanding and recognition of different STIs and their modality of spread. Research from low- and middle-income countries (LMICs) indicates limited knowledge and awareness of STIs [40]. Furthermore, literature from sub-Saharan Africa demonstrates that STIs are often discussed in the context of HIV, with HIV being the most well-known STI [40, 41]. It is, therefore, possible that individuals assume that a pill that protects against HIV will also provide protection against other STIs. These ideas may have been reinforced by radio programs (and other media) reported by our policy makers, which state that PrEP is intended for those who are unable to use condoms consistently. Other work has also reported confusion by PrEP end-users at the prospect of concurrent PrEP and condom use [42]. We, therefore, recommend public health communication campaigns targeted specifically at addressing the misconceptions found in this study to support a successful PrEP delivery strategy in Lesotho and similar settings.

In addition to the misconception that PrEP protects against STIs other than HIV and pregnancy, our respondents reported a common conflation of ART and PrEP. Addressing this

misconception is important because it may lead to increased stigma around the use of PrEP, which in turn may negatively affect PrEP use and disclosure [21, 32, 43]. In some studies, stigma has led to early discontinuation of PrEP use (29). Ways to more clearly distinguish PrEP from ART could include packaging that bears no resemblance to ART–both in the packet and the pill appearance itself (19). Additionally, tailored marketing terminology, adapted to the target audience and delivered in accessible ways, could highlight differences between ART and PrEP. Furthermore, great care is required in distinguishing conceptual differences between PrEP as a prophylactic drug for use by an HIV-negative individual, and the reduced risk of HIV transmission resulting from consistent ART use in persons living with HIV. Previous studies have demonstrated that the risk of transmission diminishes with lower viral levels [44, 45]. This concept showing undetectable levels of HIV equaling non-transmissibility of the virus (undetectable equals untransmittable; U = U) requires the HIV-positive individual to achieve viral suppression, which may take over 6 months of consistent ART use [6]. Both strategies—PrEP and U = U—are aimed at reducing risk of HIV transmission and both require adequate counseling and monitoring by health providers.

We found sources of PrEP information to be traditional–TV, radio, and education sessions at healthcare facilities–which have also been documented in other studies [23]. These traditional avenues lack interaction that would allow clients to ask questions and/or allay concerns. Having to go to a healthcare facility to be able to ask such questions may constitute an important barrier to increasing PrEP uptake. As more people in sub-Saharan Africa gain access to smart phones and internet services [46], a digital messaging strategy or social media engagement may offer an avenue for more targeted and interactive PrEP promotion. Future research to explore these digital modalities for communication around PrEP programming is necessary.

Our study's strength included the wide array of participants recruited from multiple levels of the health system. This feature allowed us to view PrEP knowledge and attitudes holistically, and triangulate information across our respondent groups. However, our research also has several limitations. First, given the sensitive subject matter, it is possible that social desirability bias may have led PrEP users to provide the information that they thought healthcare providers or the general public would approve. To mitigate this bias, data collectors invested in rapport building prior to initiating interviews, and reassured respondents that their responses would not affect the care they receive. The respondents were also assured that no identifiable information would be reported in subsequent publications. Confidentiality was particularly salient for policy makers and implementing partners, who were identified in collaboration with Ministry of Health officials. Given the high-ranking positions policy makers and implementing partners included in the study occupied, and that this work was being done, in part, to support the implementation of the national PrEP program, it is unlikely that the involvement of Ministry of Health officials hindered the respondents' full engagement in the interviews. Second, we experienced substantial challenges in accessing adults who were eligible to use PrEP but declined to use it, because this information was neither being recorded nor tracked at healthcare facilities during the time of the study. Third, our study did not systematically assess PrEP knowledge. Some other studies have found that despite high reported awareness about PrEP among healthcare providers, granular knowledge on the intricacies of the PrEP is lacking [47]. Finally, although this study included respondents across several ages, we encourage future research that more pointedly captures perspectives from those who identify as male or non-binary; men retain tremendous influence on household decision-making related to healthcare seeking and their insights thus merit pointed examination. Furthermore, MSM bear a disproportionate burden of risk relative to HIV. We hope our findings spark future research among these groups.

While our study suggests relatively high general knowledge of PrEP, we also found important misconceptions about PrEP among the general public in Lesotho. An effective PrEP communication plan is, therefore, likely to be essential for the success of Lesotho's national scale-up of PrEP. PrEP benefits expressed by our study respondents–keeping families together, allowing for conception with an HIV-positive partner while minimizing the risk of HIV infection, providing agency and control, and helping the country address HIV–can be leveraged to shape a positive message around the drug. Concurrently, such a communication strategy will need to directly address the misinformation and misconceptions identified by our respondents, including the belief that PrEP protects against STIs other than HIV, promotes promiscuity, prevents pregnancy, causes seroconversion, and provides lifelong protection from taking just one pill.

## Supporting information

**S1 Appendix. Semi-structured in-depth interview and focus group discussion guides.** (DOCX)

**S2 Appendix. Qualitative codebook.** (DOCX)

**S1 Questionnaire. Inclusivity in global research.** (DOCX)

## Author Contributions

**Conceptualization:** Pascal Geldsetzer, Joy J. Chebet, Tapiwa Tarumbiswa, Rosina Phate-Lesihla, Chivimbiso Maponga, Esther Mandara, Till Bärnighausen, Shannon A. McMahon.

**Data curation:** Joy J. Chebet.

**Formal analysis:** Joy J. Chebet.

**Funding acquisition:** Pascal Geldsetzer, Till Bärnighausen.

**Investigation:** Pascal Geldsetzer, Tapiwa Tarumbiswa, Rosina Phate-Lesihla, Shannon A. McMahon.

**Project administration:** Pascal Geldsetzer, Joy J. Chebet, Tapiwa Tarumbiswa, Rosina Phate-Lesihla, Chivimbiso Maponga, Esther Mandara, Till Bärnighausen, Shannon A. McMahon.

**Supervision:** Pascal Geldsetzer, Joy J. Chebet, Chivimbiso Maponga, Esther Mandara, Shannon A. McMahon.

**Writing – original draft:** Pascal Geldsetzer, Joy J. Chebet.

**Writing – review & editing:** Pascal Geldsetzer, Joy J. Chebet, Tapiwa Tarumbiswa, Rosina Phate-Lesihla, Chivimbiso Maponga, Esther Mandara, Till Bärnighausen, Shannon A. McMahon.

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
