## [Decision Letter · Decision Letter 0]

19 Apr 2022

PGPH-D-22-00344

Knowledge and attitudes about HIV pre-exposure prophylaxis: Evidence from in-depth interviews and focus group discussions with policy makers, healthcare providers, and end-users in Lesotho

Dear Dr. Geldsetzer,

Thank you for submitting your manuscript to PLOS Global Public Health. After careful consideration, we feel that it has merit but does not fully meet PLOS Global Public Health’s publication criteria as it currently stands. Therefore, we invite you to submit a revised version of the manuscript that addresses the points raised during the review process.

Please address all comments raised by the reviewers.

Please submit your revised manuscript by . If you will need more time than this to complete your revisions, please reply to this message or contact the journal office at globalpubhealth@plos.org. Please include the following items when submitting your revised manuscript:

We look forward to receiving your revised manuscript.

Kind regards,

Tsitsi G. Monera-Penduka

Academic Editor

Journal Requirements:

2.  Your co-authors:

Tapiwa Tarumbiswa -drtarumbiswa@gmail.com

Chivimbiso Maponga -cmaponga@clintonhealthaccess.org

Till Bärnighausen -till.baernighausen@uni-heidelberg.de

Shannon A. McMahon -shannon.aleece@gmail.com

,have not confirmed authorship of the manuscript. We have resent them the authorship confirmation email; however please check that the above email address for them is correct and follow up personally to ensure they confirm. 

Please note that we cannot proceed your manuscript  until we have received confirmations from all co-authors.

Additional Editor Comments (if provided):

This is a well written manuscript. Please address comments from reviewers.

Reviewers' comments:

Reviewer's Responses to Questions

**Comments to the Author**

1. Does this manuscript meet PLOS Global Public Health’s publication criteria? Is the manuscript technically sound, and do the data support the conclusions? The manuscript must describe methodologically and ethically rigorous research with conclusions that are appropriately drawn based on the data presented.

Reviewer #1: Yes

Reviewer #2: Yes

Reviewer #3: Yes

2. Has the statistical analysis been performed appropriately and rigorously?

Reviewer #1: N/A

Reviewer #2: N/A

Reviewer #3: N/A

3. Have the authors made all data underlying the findings in their manuscript fully available (please refer to the Data Availability Statement at the start of the manuscript PDF file)?

Reviewer #1: No

Reviewer #2: No

Reviewer #3: Yes

4. Is the manuscript presented in an intelligible fashion and written in standard English?

Reviewer #1: Yes

Reviewer #2: Yes

Reviewer #3: Yes

5. Review Comments to the Author

Reviewer #1: Thank you so much for letting me review this important contribution to the field of PrEP and global public health. This is a well written and interesting piece, which I am sure will contribute to the fight against HIV.

1. I have a comment regarding the sampling. The authors write that they achieved diversity among PrEP end users to represent a variety of ages etc. When I am looking into Table 1, I notice that there are important gaps in representation and this needs to be addressed in the text. Females were 69% of the users, 86% of former users and 100% of the decliners (no non-binary or other trans people?). Where are the men? If the aim is to increase uptake, one has to understand the groups who currently are not being reached. Men who have sex with men were mentioned once but looking at the Respondent HIV Risk category, there were none included? This is problematic for several reasons, the main being that MSM and MSM sex workers are usually disproportionate negatively affected by HIV also in settings with a high prevalence in the general population. Although homosexuality is not criminalised in Lesotho, there is no legal protection and stigma remains high, this is a barrier towards getting MSM/MSM sex workers into clinics for PrEP enrollments. Under “PrEP knowledge” I read that end-users cognizant that PrEP users were often MSM. At the same time, they are not part of this study? This must be discussed in Limitations.

2. Regarding the large group of former users, I would like to know why they stopped? They had positive attitudes towards PrEP but chose not to use it any more. New life choices? Desirability bias? A large group of former users had the experience of sex work. I wonder, did they stop sex work or why did they stop?

3. In the result section, I am curious how the respondents related to U=U in the context of PrEP. The respondents talk about how PrEP enables relationships with people living with HIV and that is great. But doesn’t U=U do that as well? I would like a short paragraph on that. Also, in my own previous research on PrEP, being on PrEP meant being an ally to the community of people living with HIV/AIDS. Did someone voice that? I noted the opposite, that the mix-up between ART and PrEP caused stigma.

4. As I mentioned earlier, if the desired implication is a deeper understanding of how to facilitate a broader appeal of PrEP programs, it would have been interesting to learn more about negative aspects or reservations about PrEP. It is mentioned briefly in the last paragraphs of the Results section but I would like to see a little bit more on that. Are there obsticles articulated from a structural perspective, for instance faith-based organisation or similar?

5. I noted that the reference on the efficiency of PrEP is, when writing about PrEP, a bit old (2016). Maybe a newer reference, with the focus on real-life settings.

6. Under the headline “PrEP knowledge”, the first paragraph and quotation. I am not sure how to understand the text and how the quotation underpin the text.

Reviewer #2: Abstract:

Clear and easy to comprehend.

Although this evidence: what evidence?

Intro: Prevalence data from 2018 used.

Study design: Would be nice to have added a sentence mentioning that the study was qualitative in nature.

Through discussions with the Lesotho Ministry of Health: Key informants were identified in collaboration with the Ministry of Health: Doesn’t it introduce biasness- how was it mitigated?

Data analysis: Themes were developed from detailed debriefing notes and analysis workshops conducted between interviewers, FGD moderators, and the study investigators: Who facilitated the analysis workshops? How many workshops were required? How were different themes from different individuals harmonized or agreed?

If the themes were developed from the briefing notes- how were interview transcriptions used for the themes identification?

How many key themes were generated?

Results:

Prep-knowledge:

End-users were cognizant that PrEP is an antiretroviral drug (ARV) and about the priority groups for PrEP use, including individuals in serodiscordant relationships, individuals with multiple sexual partners, and MSM: Sentence difficult to comprehend

Would be nice to quantify data on end-users like

- overwhelming number of people heard about prep from radio and television:

- to a lesser degree

- a minority of

While the researcher have provided numbers (two respondents, one users etc) in many instances.

Discussion:

Among policy makers, implementing partners, and healthcare providers, PrEP evoked a sense of hope for addressing the HIV epidemic in Lesotho: Just curious if the epidemiological data supported this since PreP was initiated in 2016 (almost 5 years ago) in 5 of 10 districts

It would be nice to have information on the need of further studies in the recommendation, since there are few studies as mentioned in the introduction.

Would be nice if the recommendation also provide more on the communication strategy details like behavior change or campaign, targeted messaging to youth, women etc- if its relevant to the study findings.

Reviewer #3: This manuscript describes results from qualitative research on Knowledge and attitudes about HIV pre-exposure prophylaxis with policy makers, healthcare providers, and end-users in Lesotho. The manuscript is timely for programming efforts in Lesotho and could serve as a model for conducting formative research around the same topic in other countries.

Generally, the manuscript is well-written and clear in both the methods and discussion. A few comments are noted:

Abstract

Nice summary of the results.

Introduction

The authors state that “Research in sub-Saharan Africa done in this area has mostly been carried out in the context of large-scale clinical trials (18–20). There is, thus, little evidence on awareness, knowledge, and perceptions of PrEP from effectiveness and implementation studies conducted in sub-Saharan Africa.” Yet they cite a multitude of studies (references 23-25, 40, 43) from the region in their discussion. I am not clear the point the authors are trying to make in the introduction.

Methods

Were all PrEP end-users adults? Perhaps it would be useful to state the eligibility criteria for participation a bit more clearly.

The authors state that “facilities were selected for the recruitment of study participants based on: 1) type of health facility: including government, private and faith-based facilities; 2) mode of PrEP delivery: community- and facility based; and 3) location: urban and rural.” It would be interesting to have this information reflected in Table 1 as well as the geographic spread of all respondents.

Also in Table 1, “number of data collection activities” is a bit strange in terms of a header – perhaps revise?

What is the difference between IDIs and semi-structured IDIs? In the narrative, the authors state that all IDIs are semi-structured.

Results

No comments; very nice reporting of qualitative results

Discussion

Discussion is very clear and organized. Additional messaging to consider would be clear articulation of the need for dual protection if not interested in getting pregnant, and clear messaging about male condom use (i.e, not needing to use condoms for disease prevention if using PrEP; need to use condoms or other contraceptive method to protect against unintended pregnancy).

6. PLOS authors have the option to publish the peer review history of their article (what does this mean?). If published, this will include your full peer review and any attached files.

**Do you want your identity to be public for this peer review?** For information about this choice, including consent withdrawal, please see our Privacy Policy.

Reviewer #1: No

Reviewer #2: **Yes: **Gaj Bahadur Gurung

Reviewer #3: No

---

## [Decision Letter · Decision Letter 1]

29 Aug 2022

Knowledge and attitudes about HIV pre-exposure prophylaxis: Evidence from in-depth interviews and focus group discussions with policy makers, healthcare providers, and end-users in Lesotho

PGPH-D-22-00344R1

Dear Dr. Geldsetzer,

We are pleased to inform you that your manuscript 'Knowledge and attitudes about HIV pre-exposure prophylaxis: Evidence from in-depth interviews and focus group discussions with policy makers, healthcare providers, and end-users in Lesotho' has been provisionally accepted for publication in PLOS Global Public Health.

Best regards,

Julia Robinson

Staff Editor

Reviewer Comments (if any, and for reference):

Reviewer's Responses to Questions

**Comments to the Author**

1. If the authors have adequately addressed your comments raised in a previous round of review and you feel that this manuscript is now acceptable for publication, you may indicate that here to bypass the “Comments to the Author” section, enter your conflict of interest statement in the “Confidential to Editor” section, and submit your "Accept" recommendation.

Reviewer #1: All comments have been addressed

Reviewer #2: (No Response)

Reviewer #3: All comments have been addressed

2. Does this manuscript meet PLOS Global Public Health’s publication criteria? Is the manuscript technically sound, and do the data support the conclusions? The manuscript must describe methodologically and ethically rigorous research with conclusions that are appropriately drawn based on the data presented.

Reviewer #1: Yes

Reviewer #2: Partly

Reviewer #3: Yes

3. Has the statistical analysis been performed appropriately and rigorously?

Reviewer #1: Yes

Reviewer #2: Yes

Reviewer #3: N/A

4. Have the authors made all data underlying the findings in their manuscript fully available (please refer to the Data Availability Statement at the start of the manuscript PDF file)?

Reviewer #1: Yes

Reviewer #2: Yes

Reviewer #3: No

5. Is the manuscript presented in an intelligible fashion and written in standard English?

Reviewer #1: Yes

Reviewer #2: No

Reviewer #3: Yes

6. Review Comments to the Author

Reviewer #1: Thank you for addressing my comments in a comprehensive way. My reccommendation is accept.

Reviewer #2: Abstract

First sentence may not be necessary. The key population is the focus on other regions such as Asia, EECA etc

Clear and easy to comprehend.

Study design:

Through discussions with the Lesotho Ministry of Health: Key informants were identified in collaboration with the Ministry of Health: Doesn’t it introduce biasness- how was it mitigated?

Snowball sampling informed later data collection, as a means of further expanding respondent groups: not clear

Semi-structured in-depth interviews and in-depth semi-structured interviews: inconsistent use of terms

Numerous long and unclear sentences- please re-write them

All IDIs and FGDs were semi-structured: repeated numerous time

The interviewers took detailed notes during the interview in case of recording failure: how many failure as it could compromise the data quality

Results:

Please shorten the quotes length in numerous instances

Recommendations to improve PrEP knowledge: should be moved to the discussion with concrete recommendations

Reviewer #3: (No Response)

7. PLOS authors have the option to publish the peer review history of their article (what does this mean?). If published, this will include your full peer review and any attached files.

**Do you want your identity to be public for this peer review?** For information about this choice, including consent withdrawal, please see our Privacy Policy.

Reviewer #1: No

Reviewer #2: No

Reviewer #3: No
